# Boron Neutron Capture Therapy and Photodynamic Therapy for High-Grade Meningiomas

**DOI:** 10.3390/cancers12051334

**Published:** 2020-05-23

**Authors:** Yukiko Nakahara, Hiroshi Ito, Jun Masuoka, Tatsuya Abe

**Affiliations:** Department of Neurosurgery, Faculty of Medicine, Saga University, Saga 840-8501, Japan; f8257@cc.saga-u.ac.jp (H.I.); masuoka@cc.saga-u.ac.jp (J.M.); abet@cc.saga-u.ac.jp (T.A.)

**Keywords:** boron neutron capture therapy, photodynamic therapy, meningioma

## Abstract

Meningiomas are the most common type of intracranial brain tumors in adults. The majority of meningiomas are benign with a low risk of recurrence after resection. However, meningiomas defined as grades II or III, according to the 2016 World Health Organization (WHO) classification, termed high-grade meningiomas, frequently recur, even after gross total resection with or without adjuvant radiotherapy. Boron neutron capture therapy (BNCT) and photodynamic therapy (PDT) are novel treatment modalities for malignant brain tumors, represented by glioblastomas. Although BNCT is based on a nuclear reaction and PDT uses a photochemical reaction, both of these therapies result in cellular damage to only the tumor cells. The aim of this literature review is to investigate the possibility and efficacy of BNCT and PDT as novel treatment modalities for high-grade meningiomas. The present review was conducted by searching PubMed and Scopus databases. The search was conducted in December 2019. Early clinical studies of BNCT have demonstrated activity for high-grade meningiomas, and a phase II clinical trial is in progress in Japan. As for PDT, studies have investigated the effect of PDT in malignant meningioma cell lines to establish PDT as a treatment for malignant meningiomas. Further laboratory research combined with proper controlled trials investigating the effects of these therapies is warranted.

## 1. Introduction

Meningiomas are the most common type of central nervous system tumors, accounting for 30% of primary intracranial tumors [1,2]. According to the 2016 revision of the World Health Organization (WHO) classification (4th edition), meningiomas are classified into three histological grades and 15 subtypes. The WHO classification is used to predict the prognosis of patients. More than 80% of meningiomas are benign and categorized as WHO grade I. WHO grade I meningiomas occur most often in women and are associated with a relatively good outcome. About 20–25% and 1–6% of meningiomas are WHO grade II and III, respectively. These high-grade meningiomas have significantly higher recurrence rates and a shorter survival time, in comparison with benign WHO grade I meningiomas. Clinically, these high-grade meningiomas remain problematic, and their management is very difficult due to limited treatment options [3]. Malignant meningioma is particularly an aggressive tumor, resulting in poor local control and overall survival.

Recent increased understanding of the molecular alterations in meningiomas has fueled extensive research efforts to develop molecularly targeted agents for treatment-refractory meningioma, but as of now, no agents have been clinically approved [4]. New modalities based on the concept of tumor selective destruction have been introduced for the treatment of brain tumors. Boron neutron capture therapy (BNCT) is a treatment modality that requires highly selective nonradioactive boron-10 (^10^B) compounds that accumulate to sufficient concentrations within a tumor, and adequate delivery of thermal neutrons to the target location to induce tumor cell death [5,6,7]. Photodynamic therapy (PDT) is a treatment modality that involves the selective uptake of a photosensitizer (PS) by tumor cells, followed by irradiation of the tumor with light of the appropriate wavelength to excite the PS [8,9]. These therapeutic modalities are distinct, in that each strategy uses unique reagents, irradiation methodologies, and resulting cytotoxic molecules. However, the common therapeutic concept shared by BNCT and PDT is that these therapies induce the accumulation of a substance in certain tumor cells and spare normal brain tissue. BNCT and PDT have been commonly performed to treat malignant brain tumors represented by gliomas; however, few reports have demonstrated that these treatments are indicated for meningiomas. We therefore consider that a comprehensive overview of the current state and discussion of challenges in advancing BNCT and PDT is timely and meaningful.

In the present review, we examine the literature on BNCT and PDT in the field of brain tumors and discuss potential future development of both therapies.

## 2. Results and Discussion

### 2.1. Boron Neutron Capture Therapy

#### 2.1.1. Background of BNCT

Contrary to conventional radiation therapy, BNCT is essentially a biologically targeted radiation therapy that selectively kills tumor cells and avoids damage to the surrounding normal tissue. Based on this mechanism, BNCT has been proposed as an innovative and ideal radiation therapy for many types of cancer since 1950s [10]. BNCT is based on the nuclear reaction that occurs when nonradioactive ^10^B is irradiated with low-energy thermal neutrons. When ^10^B captures a neutron, however, the unstable isotope ^11^B is created, which subsequently undergoes instantaneous nuclear fission into recoiling lithium-7 (^7^Li) nuclei and high-energy alpha particles (^4^He). These ^7^Li nuclei and alpha particles (^4^He) are high linear energy transfer (LET) products, which deposit their energies in a range of 5–9 µm [11]. The average path of these particles in biological systems is shorter than the cell diameter; hence, the harmful effects of these compounds with high LET products are limited to cells containing boron (Figure 1). If the efficiency of BNCT is restricted by thermal neutrons, methods of converting light water to heavy water—up to 23%—can be useful for increasing the maximum therapeutic depth. Therefore, it is possible to selectively irradiate cancer cells that have taken up a sufficient amount of ^10^B, while simultaneously sparing normal cells [10,12].

#### 2.1.2. Boron Compounds

Achieving a successful therapy requires highly selective boron compounds that have been delivered to tumor cells. It is necessary to understand the properties of boron compounds and the conditions in which they exist. Boron compounds have to be selectively delivered and remain in the tumor tissue during the appropriate time of irradiation. The tumor-to-blood ratio should be maximal; additionally, the boron concentration of the tumor tissue should be at least three times greater than that in the normal brain [13]. In the 1960s, two boron compounds emerged from investigations of hundreds of low-molecular-weight boron-containing chemicals. One was L-4-dihydroxy-borylphenylalanine, which is called boronophenylalanine (BPA), and the other was sodium mercaptoundecahydro*-closo-*dodecaborate (N_a2_B_12_H_11_SH), known as sulfhydryl borane (BSH), based on a newly discovered polyhedral borane anion. These drugs had low toxicity and both the tumor/brain and tumor/blood boron ratios were greater than 1. Therefore, the majority of clinical trials to date used either BPA or BSH [12].

BPA is transported across the blood brain barrier into the normal brain tissue, and the tumor uptake of BPA is related to an elevated rate of amino acid transport at the tumor cell membrane. The concentration of BPA in intracranial tumors is two to four times greater than that in the normal brain or blood. It is very soluble, and its toxicity is low when administered 200–400 mg/kg by intravenous infusion over approximately 2 hours. Irradiation starts approximately one hour after the BPA infusion has ended [12].

In contrast, BSH does not cross the blood brain barrier in the normal brain. Accumulation of BSH in the tumor tissue results from the passive diffusion from blood through an impaired blood brain barrier. BSH is administered to patients undergoing craniotomy, and its concentration is measured in the tumor, blood, and normal brain tissue. BSH was taken specifically in the brain tumors after the first 18 hours; it is virtually excluded from normal brain tissue [14,15].

#### 2.1.3. Boron Uptake Imaging with PET

BPA is a member of the phenylalanine family and delivers boron to tumor cells. Tumor cells during the S-phase of deoxyribonucleic acid (DNA) synthesis are more likely to actively adsorb essential amino acid congeners. Therefore, BPA accumulation within tumor cells increases during the cell cycle [16]. ^18^F-BPA formulated by electrophilic fluorination with fructose was studied in vivo with positron emission tomography (PET) [17]. ^18^F-BPA-PET has been shown to allow the measurement of BPA accumulation and distribution in the tumors, and prediction of the effect of BNCT before irradiation without craniotomy [17,18]. However, the synthesis of ^18^F-BPA is very complicated. ^11^C methionine (^11^C MET) is commonly used as a positron labeled amino acid probe and is easily available in clinical facilities. Nariai et al. (2009) compared the uptake of ^18^F-BPA and ^11^C-MET in malignant gliomas after infusion of those drugs at the same time [19]. A linear correlation was found between the tumor-to-normal brain ratio of ^18^F-BPA and ^11^C-MET. This study suggested that ^11^C-MET PET imaging might be more useful for screening candidates for BNCT than ^18^F-BPA.

#### 2.1.4. BNCT for High-Grade Meningiomas

We conducted a literature search across the PubMed and Scopus databases using the key terms “boron neutron capture therapy” and “meningioma” and identified 13 published reviews, 10 original articles, and five case reports. Table 1 shows the list of original articles and case reports.

The development of boron agents was important for the success of BNCT. The biodistribution of the boron agent, BSH, was first examined in eight dogs with meningiomas in 1994 [14]. In 1995, the measurement of BSH concentration was conducted in patients with intracranial brain tumors—including meningiomas. Intraoperative sampling of blood, normal brain tissue, and the tumor was performed after the injection of BSH, and meningiomas showed boron concentrations comparable with high-grade intracranial tumors (glioblastoma, anaplastic astrocytoma, medulloblastoma, and metastasis) with a tumor-to-blood ratio above 1 [15]. Furthermore, ^18^F-BPA-PET was shown to be useful to predict the accumulation of BPA in meningiomas [31]. Several authors reported that the tumor-to-blood ratio in meningiomas was between 1.5 and 5.0 by quantitative uptake of ^18^F-BPA-PET [6,18,21,29].

The first case of a recurrent papillary meningioma, categorized as WHO grade III and treated by BNCT, was reported in 2006 [30]. Prior to carrying out BNCT, the authors performed ^18^F-BPA-PET to the patient, in order to assess the boron concentration in the tumor. The tumor/ normal brain tissue ratio revealed a high uptake of ^18^F-BPA enough to apply BNCT for the patient. BNCT was then administered to the patient in a manner similar to their previous patients with high-grade gliomas—500 mg/kg of BPA and 5 g of BSH. Following this report, other reports of BNCT for high-grade meningiomas with a WHO grade of II and III, have emerged from a number of BNCT groups [20,25,26,28]. Most of the articles reporting the clinical outcome of BNCT are from Japan as shown in Table 1. The recent publication in 2018 summarized a relatively large series of cases of recurrent high-grade meningiomas (*n* = 33) treated at Osaka Medical College between 2005 and 2014 [20,25,28]. A mean volume reduction of 64.5% was achieved after 2 months of BNCT [25]. Although the treatable depth has not been established in BNCT, from the perspective of the attenuation of the neutron flux, this article demonstrated a new knowledge that suppression of meningiomas located in the skull-base was comparable to those in the cranial surface [20]. In addition, the median survival time after BNCT and being diagnosed as high-grade were 24.6 and 67.5 months, respectively, regardless of whether the tumor was located in the skull-base or not [20]. Overall, these early stage clinical investigations of BNCT have demonstrated an encouraging sign of anti-tumor activity against high-grade meningioma. Some cases of meningiomas treated by BNCT showed a transient increase in tumor volume in image findings immediately after treatment [27]. As with conventional radiotherapies, treatment with BNCT resulted in difficulties with differentiating actual tumor recurrence and pseudo-progression with necrosis. ^18^F-BPA-PET, using indices of several parameters (standard uptake value (SUV) mean, SUV max, metabolic tumor volume), indicated the possibility of differentiating tumor recurrence from necrosis [21].

#### 2.1.5. Adverse Events and Limitations of BNCT

Clinical studies of BNCT for high-grade meningiomas have demonstrated its ability to control tumors locally. However, recurrences after BNCT occur, such as intracranial recurrence outside the irradiation field, cerebrospinal fluid (CSF) dissemination, and systemic metastasis, including lung, bone, and liver. BNCT may not be suitable for deep tumors, as the attempt to apply a sufficient absorbed dose to deep-seated tumors could increase the dose to the normal brain. The treatable depth limit in BNCT remains undetermined [20]. To address this problem, new methods and techniques have been developed; for example, the CSF in the cavity of the tumor resection is instead replaced by air via the inserted Ommaya reservoir [32]. High-LET radiotherapy, such as BNCT, results in radiation induced brain edema, necrosis, and pseudoprogression. It should be recognized that these phenomena occur not only with malignant gliomas, but also with high-grade meningiomas after BNCT [27]. Because most patients with a high-grade meningioma have a history of radiation treatment, such as conventional and or stereotactic radiosurgery, particular attention to these adverse events should be needed. Psudoprogression was observed in three of 13 malignant meningiomas within 3 months after BNCT [27]. Pseudoprogression after high-LET radiotherapy manifests an increased enhanced volume of the tumor in imaging. ^18^F-BPA-PET is useful to distinguish radiation induced necrosis from recurrence [33]. Endothelial cell damage resulting in the disruption of the blood-brain barrier and edema is presumed to be induced by BNCT. Vascular endothelial growth factor (VEGF) is also associated with radiation necrosis and edema; thus, anti-VEGF antibodies might be effective [10,31,34].

#### 2.1.6. Future Prospect of BNCT

In the last few decades, one of the key factors for success with BNCT was the development of boron carriers, which could work safely and effectively in humans. There has been success utilizing BHS and BPA. A second important aspect is the availability of a reliable neutron source. Prior to 2012, all clinical irradiations using BNCT were carried out at nuclear reactors—places which required large areas and structures. Over the past 60 years, only 15 neutron facilities, all nuclear research reactors, have been available for a limited number of investigator groups to perform BNCT for patients with a variety of cancer types. Recently, accelerator-based neutron sources have been developed and proposed in the hospital setting [6]. The accelerator-based neutron source is more compact and less expensive than a reactor.

Supported by the Japanese government, the Japanese researchers and companies have collaborated to develop BNCT therapeutic systems. The intravenous drip bag of boropharan (^10^B) referred to as Steboronine^®^ was created by STELLA PHARMA CORPORATION (Shinagawaku, Tokyo). Furthermore, TOKYO-Sumitomo Heavy Industries, Ltd. developed a BNCT system using an accelerator (cyclotron). A Japanese phase II study using these new drug and device was conducted in 21 patients with locally unresectable recurrent squamous or non-squamous cell carcinomas of the head and neck. Overall response rate at 90 days after neutron irradiation was 71.4% (95% CI: 47.8–88.7%), and deaths or discontinued cases were not observed during the primary endpoint evaluation period (90 days after neutron irradiation) [35]. As a result of this clinical trial, boropharan (Steboronine^®^) and the accelerator-based BNCT system were recently approved by the Ministry of Health, Labor and Welfare of Japan in 2020. These are the world’s first medical boron drug and devices for BNCT. A phase II clinical trial of accelerator-based BNCT for high-grade meningioma initiated in April 2019, and is ongoing in Japan (jRCT2051190044; https://jrct.niph.go.jp/). The study was planned to enroll 18 patients with recurrent high-grade meningioma (WHO grade II or III) and are divided into two groups by stratified and blocking randomization, with 12 cases being treated with BNCT and six cases in the control group with best supportive care. The primary endpoint is progression-free survival.

It is expected that therapeutically accelerator-based BNCT will be comparable to nuclear reactor-based BNCT. As a result of the compact size and relatively low costs (about USD 30 million to USD 40 million), accelerator-based neutron sources have been proposed in the hospital setting. The demonstration of feasibility and efficacy of accelerator-based BNCT in clinical trials for tumors including high-grade meningiomas could increase the recognition of its practicality, and broaden the availability of the system worldwide, opening up potential opportunities for global radiation oncology groups to initiate clinical studies of BNCT.

### 2.2. Photodynamic Therapy

#### 2.2.1. Background on PDT

PDT was originally developed as an additional therapy that enhanced the effect of surgery nearly 25 years ago [36]. With this therapy, the patient receives a drug called a photosensitizer (PS), which makes target tissues sensitive to a certain light. PDT consists of three essential components: a photosensitizer (PS), light, and oxygen. Although each component is not toxic, together they initiate a photochemical reaction. The PS localized in target cells, absorbs the photon when subjected to a certain wavelength of light. This photon then triggers two types of photochemical reactions. The first type of interaction occurs directly with the molecules of the substrate and leads to the formation of free radicals. In the second type of interaction, an excited PS interacts with oxygen to form singlet oxygen. Singlet oxygen is cytotoxic to living cells, and it only exists for < 4 µs and migrates for a maximum of 1 micrometer (Figure 2).

PDT has an antitumor effect derived from three mechanisms: (i) direct cytotoxic effects by the generation of reactive oxygen species in the cytoplasm or mitochondria; (ii) damage to the tumor vasculature, resulting in ischemia and necrosis; and (iii) induction of a robust inflammatory reaction activating immune cells and cytokines. Based on these mechanisms, PDT is a biologically targeted therapy for tumor cells. Major adverse events have been reported with PDT of central nervous system tumors, such as cerebral edema, increased intracranial pressure, hypersensitivity reaction, and skin photosensitization.

#### 2.2.2. Photosensitizers (PSs) 

Porphyrin photosensitizers for antitumor therapy have been used on patients with generally encouraging results [36]. The selectivity of PDT is derived from the ability of the PS to localize in tumor cells and the precise delivery of light to the treated site. The required features of PSs are increasing, and the selective uptake into tumor cells makes the treatment specific. Important characteristics of an ideal PS include peak activation at 650–800 nm and a single component compound that is systemically non-toxic and water-soluble. It is necessary for PSs to be rapidly excreted, resulting in a reduction in skin phototoxicity, and transferred across an intact blood-brain barrier, so as to reach tumor cells without entering surrounding normal brain tissue. Most of the PSs are based on a tetrapyrrole structure, similar to that of the protoporphyrin (Figure 3). Several PSs have been used, including a hematoporphyrin derivative (HPD) [8,37,38,39,40,41,42,43], porfimer sodium (Photofrin) [9,44], 5-aminolevulic acid (5-ALA) [45,46,47,48,49,50,51,52,53,54,55,56,57], m-tetrahydroxyphenylchlorin (mTHPC, temopofin), benzoporphyrin derivative monoacids ring A (BPD-MA), and mono-l-aspartyl chlorin(e6) (NPe6, talaporfin sodium) [20,58] (Table 2). Most of the PSs are based on a tetrapyrrole structure, similar to that of the protoporphyrin (Figure 3). Several PSs have been used, including a hematoporphyrin derivative (HPD) [8,37,38,39,40,41,42,43], porfimer sodium (Photofrin) [9,44], 5-aminolevulic acid (5-ALA) [45,46,47,48,49,50,51,52,53,54,55,56,57], m-tetrahydroxyphenylchlorin (mTHPC, temopofin), benzoporphyrin derivative monoacids ring A (BPD-MA), and mono-l-aspartyl chlorin(e6) (NPe6, talaporfin sodium) [20,58] (Table 2).

#### 2.2.3. 5-Aminolevulic Acid Based PDT (5-ALA-PDT)

##### Mechanism and Fluorescence of 5-ALA

5-ALA is a precursor in the cellular heme biosynthesis pathway, and is formed in the mitochondria by the activity of the ALA synthetase. Exogenous 5-ALA enters the cytosol and bypasses the negative feedback mechanism, resulting in an increase in the fluorescent molecule protoporphyrin IX (PpIX). While PpIX accumulation in normal brain tissue is limited, it occurs in tumors, depending on the tumor cell type. Many tumors have a more efficient method for the synthesis of porphyrins. Alterations of the heme biosynthetic pathway, such as cellular uptake of 5-ALA, increases PpIX and leads to its accumulation within tumor cells. Higher cellular PpIX concentrations lead to a greater fluorescence intensity. The PpIX is excited by blue light at a wavelength between 375–440 nm, and then emits red light to photobleached PpIX and other porphyrins, allowing medical utility in both diagnosis and therapy.

##### Application of 5-ALA-PDT to Meningiomas

The literature search was performed in the PubMed and Scopus databases using the key terms “photodynamic therapy” and “meningioma” in December 2019. Articles studying PDT for meningiomas are listed in Table 3.

The use of 5-ALA for assisting the resection of high-grade gliomas (HGGs) was first published in 1998 [65]. The purpose of the use of 5-ALA was to clarify and visualize the borders between tumor and normal brain tissue in the surgical field. Since then, 5-ALA has become widely used for photodynamic diagnosis (PDD) during glioma resection surgery [66]. Over 80% of meningiomas can be visualized using fluorescence well after 5-ALA administration. In studies of malignant tumors represented by HGGs, 5-ALA-induced PpIX accumulation was correlated with the degree of malignancy. On the other hand, in meningiomas there is no correlation between pathological grade or proliferation and fluorescence intensity [49]. In addition, fluorescence intensity of the meningioma shows intratumoral heterogeneity. Previous in vitro studies indicated that PpIX production in meningioma cells was much lower than that in glioma cells, suggesting that cells with a rapid turnover produce PpIX [45,67]. There were also differences in intracellular PpIX concentrations among meningioma cell lines. For PDT using 5-ALA, one of the newly found properties of PpIX was to produce cytotoxic singlet oxygen radicals [49]. As 5-ALA-mediated PpIX tends not to accumulate in the cell nuclei, but instead in the mitochondria, PDT does not cause DNA damage [45,51,52]. Based on these data, meningiomas were not considered to be a suitable target for 5-ALA-PDT. The exact mechanisms of intratumoral heterogeneity of PpIX concentrations in meningioma are not elucidated. Ferrochelatase (FECH) in the mitochondria is the terminal enzyme of the heme biosynthetic pathway. FECH activity has long been considered a major contributor to selective PpIX accumulation in tumor cells [4,50]. If agents that inhibit FCH activity are administered before 5-ALA, it is possible to increase the concentration of PpIX in meningioma cells, resulting in an effective 5-ALA-based PDT. Similar to this concept, some in vitro studies showed that the efficacy of 5-ALA-PDT was enhanced by several anti-cancer drugs, such as gefitinib and antibiotic agents, and ciprofloxacin using meningioma cell lines [53,56].

#### 2.2.4. Talaporfin Sodium Based PDT (TS-PDT)

##### Development of Talaporfin Sodium as a Photosensitizer

In the 1970s and 1980s, the selective photodegradation of malignant tumors by HPD was proven to be a promising therapeutic modality in the treatment of cancer. However, the lack of understanding of the mechanisms for the active components, which depend on HPD uptake and retention, led to side effects in normal tissue such as the skin, liver, spleen, and kidney, which persisted up to 4 to 6 weeks after HPD administration. In particular, the first generation PSs represented by HPD had a major problem with drug-induced sensitivity to sunlight. The patients who received the PDT had to avoid sunlight for a long period. To resolve the problems of a relatively weak porphyrin absorption and inefficient phototoxicity at 630 nm, considerable efforts have been devoted to developing a new and more efficient tumor localizing PS for PDT.

The chlorins are known to have strong absorption bands with high molar extinction coefficients at wavelengths greater than 650 nm. One of the chlorins, mono-l-aspartyl chlorin (e6), also called talaporfin, has a significant absorption band located at 664 nm, and is a relatively novel PS developed for PDT. In 1987, Nelson et al. studied the photosensitizing potential of talaporfin, and demonstrated the first result of long-term survival by PDT in a subcutaneous animal model. They also reported that animals who received talaporfin and light stimulation did not show any deleterious skin reactions [68]. In Japan, talaporfin was approved for clinical use, with a semiconductor laser, as a PS of PDT for early-stage lung cancers in 2003.

##### Application of TS-PDT for Meningiomas

TS-PDT was first performed for 14 patients with malignant gliomas between 2006 to 2008 by Akimoto et al. They administered talaporfin and applied intraoperative irradiation of the walls of the resection cavity at a 664 nm wavelength [69]. A phase II study of TS-PDT using the same protocol reported that the overall survival in newly diagnosed glioblastomas was 24.8 months [70]. PDT involving talaporfin as a PS, and a diode laser device was approved for the treatment of primary malignant brain tumors in Japan in 2013. This approval prompted research to investigate TS-PDT as a possible treatment for brain tumors, with various pathological and histological diagnoses. However, currently, clinical application of TS-PDT for meningioma has not yet been reported. 

With medical insurance coverage of PDT for a range of primary brain tumors in Japan, TS-PDT can be applied not only to HGG, but also to high-grade meningiomas. Assisted by the reduction of associated costs (to about USD 80,000), installation of the PDT system has now expanded to about 30 hospitals in Japan (as of the end of March 2020), enabling clinical TS-PDT of tumors. Clinicians at those institutions have begun to apply TS-PDT to patients with high-grade meningiomas (per personal communications). At this stage, it is too early to know whether TS-PDT is effective for high-grade meningiomas, and a longer observation time will be needed to assess objective outcomes. However, the accumulation of clinical data can provide support for TS-PDT, so that its application to high-grade meningiomas becomes globally available in the near future.

Basic research, such as in vitro and animal model studies, could support the clinical effect of TS-PDT. Recently, a few studies have investigated the effects of TS-PDT in malignant meningioma cell lines to establish TS-PDT as a treatment for malignant meningiomas (Table 3). The mRNA expression of heme oxygenase-1 (HO-1), which is a stress response protein, was increased after TS-PDT. In addition, TS-PDT with the treatment of zinc protoporphyrin IX, one of the HO-1 inhibitors, increased the cytotoxic effect in meningioma cells. These data suggest that HO-1 may be involved in the resistance of TS-PDT in meningiomas [59]. Another study demonstrated that apoptotic and necrotic cell death occurred in human malignant meningioma cell lines after treatment with TS-PDT; this was found to be dose- and time-dependent. However, the sensitivity to TS-PDT in meningioma cells was lower, when compared with that in glioma cells [58]. Animal studies of TS-PDT have not been published. Investigation of the mechanisms, including the uptake and retention of talaporfin in tumor cells, may contribute to the improvement of the efficacy of TS-PDT for meningiomas.

##### Adverse Events and Limitations of TS-PDT

The efficacy of TS-PDT in treating tumors depends on the amount of TS accumulation. The cellular damage induced by TS-PDT has been assumed to be a result of vascular occlusion [71]. Based on the fact that TS binds to plasma albumin, the high concentration of TS in vessels leads to a risk of cerebrovascular occlusion that results in ischemic damage. In TS-PDT, intraoperative irradiation is applied to the surface of the tumor resection by a semiconductor laser. The penetration depth of a laser is limited to only 2.5–5 mm; therefore, PDT is not considered suitable for large tumors [69].

#### 2.2.5. Future Prospect of PDT

The future of PDT depends on the development of a generation of newly synthesized PSs. There is a need for a new PS that has more specific accumulation in different types of tumor cells, especially tumor cells with intermediate proliferation rate. In addition, various strategies are considered in order to increase the efficacy of PDT. As a new treatment approach, the combination of PDT with an application of the drug delivery system is being considered, and with various inhibitors or chemotherapeutic agents. In the field of light delivery systems, the development of in vivo fluorescence optic technology may potentially broaden the practical clinical application of PDT for tumors. For example, operative endoscopes, modified and equipped with fluorescence and laser, can be a useful tool for obtaining a tumor biopsy, as well as for applying PDT at locations difficult for microsurgical approach, such as intra-ventricular or trans-nasal surgeries.

## 3. Literature Analysis Methods

The present review was conducted by searching the PubMed and Scopus databases using the key terms “boron neutron capture therapy” or “photodynamic therapy” and “meningioma”. The search was conducted in December 2019. Publicly available, peer reviewed literature written in Japanese was included. The literature was systematically retrieved using the Preferred Reporting Items for Systematic Reviews and Meta-Analyses (PRISM) guidelines, however, relevant data was extracted to help readers’ understanding.

## 4. Conclusions

The present review summarizes the use of BNCT and PDT as treatment options in the field of brain tumors, focusing on high-grade meningioma. We believe that BNCT and PDT might offer a viable alternative to standard therapies in the treatment of malignant meningiomas. In general, because a novel modality is usually first applied to the most malignant tumors, such as glioblastomas, it will take time to apply it to tumors with intermediate to rapid growth, such as meningiomas. To demonstrate the effectiveness of BNCT or PDT in the treatment of meningioma patients, more laboratory research combined with properly controlled trials investigating the effects of therapies are necessary. In the future, there is a possibility that BNCT and PDT could be combined to treat malignant tumors, if a synthesized boronated porphyrin is successfully developed and the compound is validated to serve as a dual sensitizer for BNCT and PDT.

## Figures and Tables

**Figure 1 cancers-12-01334-f001:**
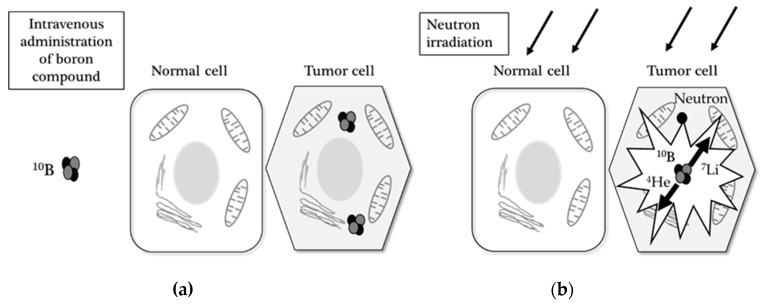
The principle of boron neutron capture therapy (BNCT): (**a**) a boron-10-labeled compound (^10^B) is administered by intravenous injection and taken up specifically in the tumor cell (the left; hexagonal cells), but not in the normal cell (the right; squared cell); (**b**) following neutron irradiation, the absorption of a neutron converts ^10^B to highly reactive alpha (^4^He) and ^7^Li particles, which results in selective tumor damage.

**Figure 2 cancers-12-01334-f002:**
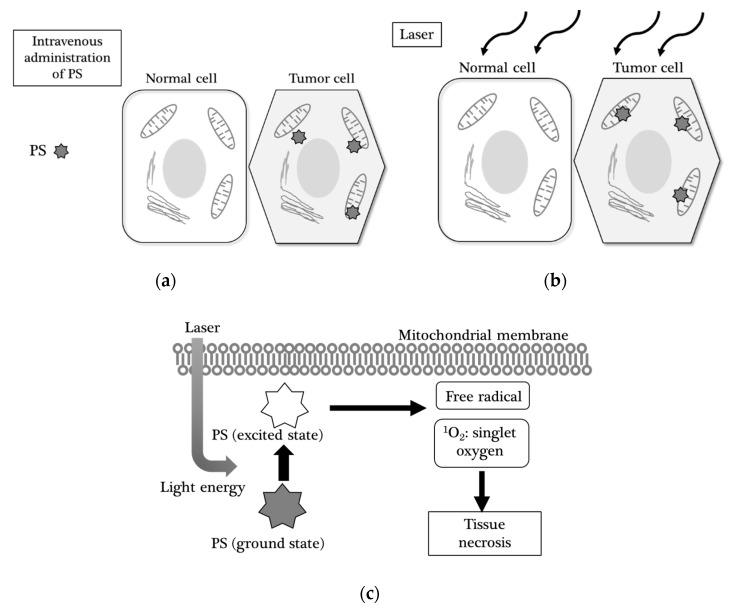
Mechanism of photodynamic therapy (PDT): (**a**) The photosensitizer (PS) is administered by intravenous injection and taken up specifically by the tumor cell (the hexagonal cell) and localizes in the mitochondria, but not in the normal cell (the squared cell); (**b**) Following irradiation of laser with a certain wavelength, the PS is excited; (**c**) in the mitochondria, an interaction occurs with reactive oxygen species including H_2_O_2_, OH^-^, leading to the formation of free radical and singlet oxygen (^1^O_2_).

**Figure 3 cancers-12-01334-f003:**
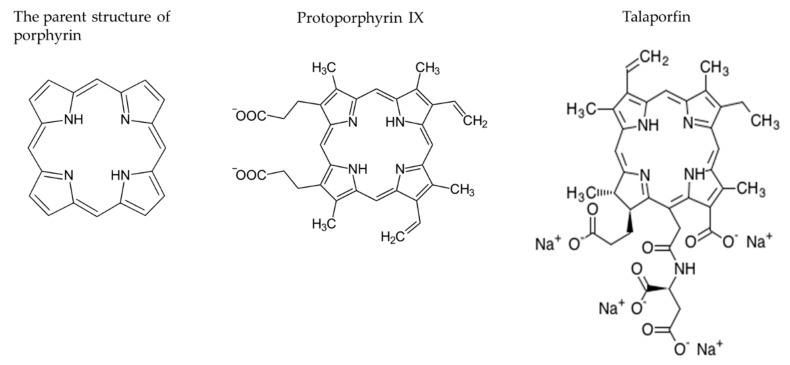
Structures of representative PSs: Porphyrins are a group of heterocyclic macrocycle organic compounds, composed of four modified pyrrole subunits interconnected at their α carbon atoms via methine bridges.

**Table 1 cancers-12-01334-t001:** Representative clinical reports and studies of BNCT for meningiomas.

Authors	Country	Published Year	No. of Patient with a Meningioma (WHO Grade)	Boron Compound	Research Target or Content	Reference
**Takeuchi, et al.**	Japan	2018	33 (12 grade II, 21 grade III)	BPA, BSH, ^18^F-BPA	Patients with skull base HGMs	[20]
**Beshr, et al.**	Japan	2018	1 (ND)	^18^F-BPA	^18^F-BPA -PET	[21]
**Kulvik, et al.**	Finland	2015	3 (3 grade I)	BPA	Biodistribution of boron	[22]
**Kageji, et al.**	Japan	2015	1 (ND)	BPA, BSH, ^18^F-BPA	Radiation-induced meningioma after BNCT	[23]
**Kawaji, et al.**	Japan	2014	1 (grade III)	BPA, BSH, ^18^F-BPA	An autopsy case after BNT	[24]
**Kawabata, et al.**	Japan	2013	20 (4 grade II, 16 grade III)	BPA, BSH, ^18^F-BPA	Patients with recurrent HGMs	[25]
**Aiyama, et al.**	Japan	2011	1 (grade III)	BPA, ^18^F-BPA	A case of MM	[26]
**Miyatake, et al.**	Japan	2009	13 (MM)	BPA, BSH, ^18^F-BPA	Pseudoprogression in patients after BNCT	[27]
**Miyatake, et al.**	Japan	2007	7 (1 grade II, 6 grade III)	BPA, BSH, ^18^F-BPA	Patients with HGMs	[28]
**Stenstam, et al.**	Sweden	2007	3 (1 grade II, 2 grade III)	BPA	Accumulation of boron	[29]
**Havu-Aurén, et al.**	Finland	2007	4 (ND)	^18^F-BPA	^18^F-BPA -PET	[18]
**Tamura, et al.**	Japan	2006	1 (1 grade III)	BPA, BSH, ^18^F-BPA	The first case of MM treated by BNCT	[30]
**Miyatake, et al.**	Japan	2006	11 (11 grade III)	BPA, BSH, ^18^F-BPA	Patients with MM	[31]
**Stragliotto, et al.**	Switzerland	1995	14 (ND)	BSH	Biodistribution of boron	[15]
**Kraft, et al.**	USA	1994	8 animal models (ND)	BSH	Biodistribution of boron	[14]

BPA, boronophenylalanine; BSH, sulfhydryl borane; ^18^F-BPA, 18F-boronophenylalanine; HGM, high-grade meningioma; PET, positron emission tomography; MM, malignant meningioma; U.S.A., United States of America.

**Table 2 cancers-12-01334-t002:** Photosensitizers (PSs) clinically approved for photodynamic therapy (PDT).

PS	1st Generation	2nd Generation	
	Porfimer Sodium	5-Aminolevulinic Acid	Talaporfin Sodium
**Structure**	Porphyrin	Porphyrin precursor	Chlorin
**Laser Device**	Excimer dye laser	Semiconductor laser	Semiconductor laser
**Wavelength** (**nm**)	630	635	664
**Target Cancers**	Lung, esophagus, bile duct, bladder, ovarian, brain	Skin, bladder, esophagus, brain	Lung, esophagus, brain
**References of Studies for Meningiomas**	[9,44]	[45,46,47,48,49,50,51,52,53,54,55,56,57]	[20,58]

**Table 3 cancers-12-01334-t003:** PDT studies for meningiomas.

Type of Study	Authors	Published Year	PS	No. of Patients with a Meningioma (WHO Grade)	Cell Lines	Reference
**In vitro**	Ichikawa, et al.	2019	talaporfin		HKBMM (human MM), KMY-J (rat MM)	[58]
**Review**	Diez, et al.	2019	5-ALA			[49]
**In vitro**	Takahashi, et al.	2018	talaporfin		KMY-J (rat MM)	[59]
**In vitro**	El-Khatib, et al.	2015	5-ALA		12 (grade I), 1 (grade II) and 1 (grade III) cultured from specimen	[48]
**Review**	Ishikawa, et al.	2015	5-ALA			[47]
**Review**	Bechet, et al.	2014	All established PSs before publish date			[57]
**In vitro**	Cornelius, et al.	2014	5-ALA		KT21-MG (human MM)	[56]
**Review**	Nokes, et al.	2014	5-ALA			[55]
**Review**	Behbananinia, et al.	2013	All established PSs before publish date			[54]
**In vitro**	Sun, et al.	2013	5-ALA		IOMM-Lee (human MM)	[53]
**Review**	Colditz, et al.	2012	5-ALA			[52]
**Review**	Hefti, et al.	2012	5-ALA			[51]
**In vitro**	Hefti, et al.	2011	5-ALA		HBL-52& BEN-MEN-1 (human benign meningioma)	[50]
**Clinical**	Eljamel, et al.	2009	5-ALA	2 (ND)		[46]
**Clinical**	Muller, et al.	2006	Profimer	3 (MM)		[9]
**Clinical**	Woehlck, et al.	2003	Verteporphin	1 (MM)		[60]
**In vitro**	Tsai, et al.	1999	5-ALA		CH-157MN (benign meningioma)	[45]
**Review**	Devaux, et al.	1996	All established PSs before publish date			[61]
**In vitro**	Malham, et al.	1996	HpD, AIPC		5 cell lines cultured from specimen (WHO grade: ND)	[62]
**In vitro**	Wilson, et al.	1996	AIPC		2 cell lines cultured from specimen (WHO grade: ND)	[63]
**Clinical**	Kostron, et al.	1995	HpD	3(MM)		[43]
**Review**	Krishnamurthy, et al.	1994	HpD			[42]
**Clinical**	Origitano, et al.	1993	Profimer	1 (ND)		[44]
**In vitro**	Marks, et al.	1992	HpD		5 cell lines cultured from specimen (4: grade I, 1: grade II)	[41]
**Review**	Powers, et al.	1992	HpD			[40]
**Clinical**	Chen, et al.	1991	HpD	1 (MM)		[39]
**In vitro**	Plattner, et al.	1991	HpD		1 cell line cultured from specimen (WHO grade: ND)	[38]
**In vitro**	Steichen, et al.	1991	Tritiated-TPP		3 cell lines cultured from specimen (WHO grade: ND)	[64]
**Clinical**	Kostron, et al.	1990	HpD	1 (MM)		[8]
**Clinical**	Kostron, et al.	1988	HpD	1 (MM)		[37]

5-ALA: 5-aminolevulinic acid, MM: malignant meningioma, HpD: hematoporphyrin derivative, AIPC: aluminum phthalocyanine chloride, TTP: tetraphenylphosphonium, ND: not described.

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
