# Peer review of "Boron Neutron Capture Therapy and Photodynamic Therapy for High-Grade Meningiomas"

_cancers, 2020, doi:10.3390/cancers12051334_

Round 1

Reviewer 1 Report

The authors performed a literature review  to investigate the possibility and efficacy of boron neutron capture therapy (BNCT) and photodynamic therapy (PDT) as novel treatment modalities for high-grade meningiomas. The authors state that the outcome of clinical studies of BNCT for high-grade meningiomas demonstrated acceptable results. As for PDT, a few studies have investigated the effect of PDT in malignant meningioma cell lines.

The treatment of higher grade meningiomas is extremely challenging and the efforts to find an effective strategy for their cure have been relatively insufficient. Therefore, any study on the topic should be considered with attention. Nevertheless, I consider this narrative review devoid of any interest for many reasons.

The combination of two very different topics is confusing. The two techniques do not have anything in common except for the fact that they require to be performed after a craniotomy. So the advantage of such adjuvant intraoperative modalities is not very advantageous over a course of postoperative radiotherapy.

Systemic therapies represent the perspective for a modern, minimally invasive treatment of these tumors. Also addressing the fact that the disease is often diffused in distant meningeal areas

The use of BNCT is impractical in 99% of the world, so the clinical interest of this technique is very limited

Both techniques did not provide any sufficient evidence of efficacy, thus a review of results I devoid of any interest.

Author Response

We presented a literature review on the use of novel strategies, boron neutron capture therapy (BNCT) and photodynamic therapy (PDT), for the treatment of high grade meningiomas. As you indicated, high-grade meningioma is often systemically diffused in distant metastases. The development of systemic therapies for meningiomas is important. However, neurosurgeons must prevent local recurrences. Essentially, both of these strategies induce the accumulation of a substance in certain tumor cells and perform irradiation to selectively damage tumor cells and spare normal brain tissue. To ensure the success of both treatments, certain substances, such as boron compounds or photosensitizers, must be developed for clinical use. With regard to BNCT, the portable neutron sources are now widely available, and it is expected that the results of clinical trials will be promising in the near future. PDT using talaporfin has been applied for malignant brain tumors in Japan, including high-grade meningiomas, and our Japanese colleagues seem to think this is promising as well (per a personal communication).

We think that it is important to share these results widely with the world, and then through basic research to investigate the mechanisms underlying the uptake of these substances.

Reviewer 2 Report

Thanks for the opportunity of reading the manuscript of the review (cancers-773732) entitled “Boron Neutron Capture Therapy and Photodynamic 2 Therapy for High-Grade Meningiomas”.

The read was very interesting and falls within the remit of my research interests and past works.

The title is well formulated and reflective of the content of the review.

Abstract:

Page 1, lines 19-20, the authors write “The present review was conducted by searching the PubMed and Scopus databases”, but I believe it would be convenient to include the date range to which this search was limited so that a reader can critically and quickly assess the relevance of this publication to their interests. If no time restriction was used, it should be stated so.

Line 20-21, with regards to the sentence “BNCT for high grade meningiomas demonstrated acceptable results”, I found this rather vague and possibly the authors could illustrate concisely the stage of clinical  success of this technique, which is in my opinion better than acceptable (considering the advances in the last decade).

Introduction:

Page 1, line 30, the authors state “maximal safe resection is the treatment of choice, followed by radiotherapy”, however the difficulties in resection in such a delicate organ like the brain are not addressed (only implied) and whether radiotherapy has proved effective is not addressed at desirable extent.

Page 2, line 61-62, sentence requires a reference.

Figure 1 is unclear. Between parts a and b there is the administration of the 10B-labeled compound, but this seems to be already in the diagram in part a. Also, the caption does not indicate the difference between the squared and hexagonal diagrams of the parts of this figure. This figre requires redrawing or many clarifications as now is not self-explanatory.

Line 84, words such as closo should be in italic (i.e. closo).

Lines 88-93 require referencing.

Line 97, what does the word “captured” actually mean here? This word might be confused with the neutron capturing process.

Lines 114-122, what is the reason for having section 2.1.4? This is out of the focus of the review; the covered literature is dated and does not capture the efforts of all applications of BNCT in brain cancer in the last 50 years. The literature is too dated to be of any use in this review. Analogously, what is the point of having 2.2.4.2? This space could be more conveniently used to give depth.

Line 146, is the expression “just higher than” taken from the original source? Also, BNCT is not administered but performed.

Line 158, abbreviations such as SUV and MTV are not defined.

Line 167, the Ommaya reservoir requires a reference.

Line 168, “radiation brain swelling”: what is the meaning?

Line 170, “it should be recognised” rather than “we should recognise” + a comma after the word gliomas.

Lines 172/173, the use of first person should be avoided. This also applies to line 245 and in the entire manuscript.

Lines 182-185, portable neutron sources are now widely available, but these are mostly reported in patents. Have authors made a search of patented work?

Figure 2 (a), same comments as per Figure 1.

Lines 219, “tetrapyrrole structures, similar to that of the protoporphyrin”, wouldn’t it be a good idea to show an example of general structure?

Lines 220-223 + Table 2. Some references for readers to track the use of those named compounds shall be provided. This could be done in the table analogously to Table 3.

At times, the take on reviewing employed materials is historical and this causes a lack of depth, e.g. in section 2.2.4.1; lines 285-286, talaporfirin was approved in Japan but there is no follow up. A reader would be disappointed in not finding up-to-date information regarding these (and analogously for the other) compounds. This time of problem has a knock down effect on the resources used. In fact, I felt most of the cited literature is quite dated; this is especially true for the parts on BNCT, but equally true for the parts on PDT as just pointed out. With few exceptions, most of the references are older than 20 years. More up to date resources must be used instead so to provide a review on correspondingly recent literature.

Phtosensitisers are sometimes abbreviated as PS’S or PSs.

Lines 335-337, sentence presents some challenges from the grammatical point of view and should be reworded.

In conclusion, while the intent of the manuscript is laudable and could offer a very useful resource to researchers in the field and clinicians, it presents some room for ameliorations as described above.

Author Response

Page 1, line 20: we added the sentence “The search was conducted in December 2019.”.

Page 1, line 21-22: we added “a phase II clinical trial in Japan” to specify the context.

Page 1, line 31-32: because there was no need for this sentence in this context, we deleted it.

Page 2, line 62-63: we inserted the proper reference

Page 2, Figure 1: we revised figure 1 and clarified the context of the figure legends.

Page 3, line 86: we included “closo” in italics.

Page 3, line 90-95: we inserted the proper reference.

Page 3, line 99: we changed “captured” to “taken”.

Page 3, line 115-124, Page 8, line 260-268, and Page 10, line 318-329: These sections 2.1.4, 2.2.3.2 and 2.2.4.2 were deleted. The descriptions of each treatments for gliomas was limited to important previous results as an introduction to meningiomas.

Page 4, line 147-150: we provided the correct descriptions from the original case report.

Page 5, line 165-166: we added the abbreviation SUV and described the full terms as MVT.

Page 5, line 175: we inserted the proper reference.

Page 5, line176: we changed “radiation brain swelling” to “radiation induced brain edema”.

Page 5, line 178: we changed “we should recognize” to “it should be recognized” and added a comma after the word “gliomas”.

Page 5, line 181-182: we changed “we should pay attention to these adverse events” to “particular attention to these adverse events should be needed”.

Page 8, line 270: we removed the use of the first person in this sentence.

Page 5, line 191-195: we have known that a small accelerator-based neutron source has been produced in Japan by Sumitomo Heavy Industries, Ltd., in which a cyclotron is used to generate the protons. A sentence “Clinical studies of BNCT for high-grade meningioma have been on going in the several institutions in Japan.” was included in this paragraph.

Page 6, Figure 2: we revised figure 2 and clarified the context of the figure legends.

Page 7, line 237: we made a new figure, figure 3, showing the structure of porphyrin, protoporphyrin IX, and talaporfin.

Page 7, line 238-241 and Table 2: we inserted the proper references in the table.

Page 5, line 192-193 and Page 10, line 333-343: the current situation in Japan was additionally described. The clinical phase II trial of BNCT for high-grade meningiomas is in progress. There are no articles or case reports on TS-PDT for meningiomas. Per personal communications, some neurosurgeons have applied it and had a good impression of TS-PDT.

Page 11, line 365: photosensitizers are abbreviated as PSs.

Page 11, line 377-380: the sentence was changed to be grammatically correct.

Reviewer 3 Report

The authors present a review of the use of the literature regarding use of novel strategies for the treatment of high grade meningiomas. Boron neutron capture therapy and photodynamic therapy are being used in treatment of high grade gliomas and are being investigated for use in high grade gliomas as well. The article does a nice job of summarizing the technology and logistics involved in these treatment strategies, as well as the preliminary investigation of whether the necessary pharmaceutical agents are indeed selectively taken up to a significant degree in high grade meningiomas.

Minor revisions

  1. The authors do not discuss the outcomes of treatment with BNCT. There is a mention of “acceptable control rates”, but a discussion of this treatment options should specifically include the outcomes and indicate whether this is an improvement over standard treatment.
  2. As these two interventions, BNCT and PDT, are in very different stages of investigation, it would be helpful to state at the outset at what stage this treatment is (ie. Animal models, clinical, or in vitro).

This is a nice review of the current literature for these two therapies. It is necessary to explicitly discuss the stage at which the different modalities currently are and, for BNCT which is being used clinically, to address the patient outcomes. Without these points, the conclusion that these are viable alternatives to standard therapy appear overstated.

Author Response

We revealed a recent clinical outcome of BNCT from Osaka Medical College on page 4, line 153-160. Several articles demonstrated the clinical course after BNCT; however, cases were shared across studies. In the section on the future prospects of BNCT, it was specified that the clinical trials of BNCT for high-grade meningiomas have been ongoing in Japan, on page 1, line20-22, and page 5, line 192-193.

We explained that PDT can be clinically applied to patients with meningioma in Japan on page 10, line 333-338. However, basic research has been insufficient in investigating underlying mechanisms, and we as such emphasized the importance of in vitro, animal studies on page 10, line 344-345.

Round 2

Reviewer 1 Report

I keep on thinking that the combination of two very different topics is confusing. So I believe that the review should be limited to one topic and better to BNCT.

The use of BNCT is impractical in 99% of the world, so the clinical interest of this technique is very limited. Nevertheless, if the authors may demonstrate that BCNT will be likely widely diffused by use of LINACs, describe the "portable" systems, and survey centers where these systems are or will be soon available, the perspectives of this study may have some clinical relevance